# Detection Method of Cow Estrus Behavior in Natural Scenes Based on Improved YOLOv5

**Rong Wang** [1,2,3], **Zongzhi Gao** [4], **Qifeng Li** [2,3], **Chunjiang Zhao** [1,2,3,*], **Ronghua Gao** [2,3], **Hongming Zhang** [1], **Shuqin Li** [1] **and Lu Feng** [2,3]

1   College of Information Engineering, Northwest A&F University, Yangling, Xianyang 712100, China
2   Research Center of Information Technology, Beijing Academy of Agriculture and Forestry Sciences, Beijing 100097, China
3   National Engineering Research Center for Information Technology in Agriculture, Beijing 100097, China
4   College of Mechanical Engineering, Beijing Institute of Petrochemical Technology, Beijing 102617, China
*   Correspondence: zhaocj@nercita.org.cn; Tel.: +86-010-51503411

**Abstract:** Natural breeding scenes have the characteristics of a large number of cows, complex lighting, and a complex background environment, which presents great difficulties for the detection of dairy cow estrus behavior. However, the existing research on cow estrus behavior detection works well in ideal environments with a small number of cows and has a low inference speed and accuracy in natural scenes. To improve the inference speed and accuracy of cow estrus behavior in natural scenes, this paper proposes a cow estrus behavior detection method based on the improved YOLOv5. By improving the YOLOv5 model, it has stronger detection ability for complex environments and multi-scale objects. First, the atrous spatial pyramid pooling (ASPP) module is employed to optimize the YOLOv5l network at multiple scales, which improves the model's receptive field and ability to perceive global contextual multiscale information. Second, a cow estrus behavior detection model is constructed by combining the channel-attention mechanism and a deep-asymmetric-bottleneck module. Last, K-means clustering is performed to obtain new anchors and complete intersection over union (CIoU) is used to introduce the relative ratio between the predicted box of the cow mounting and the true box of the cow mounting to the regression box prediction function to improve the scale invariance of the model. Multiple cameras were installed in a natural breeding scene containing 200 cows to capture videos of cows mounting. A total of 2668 images were obtained from 115 videos of cow mounting events from the training set, and 675 images were obtained from 29 videos of cow mounting events from the test set. The training set is augmented by the mosaic method to increase the diversity of the dataset. The experimental results show that the average accuracy of the improved model was 94.3%, that the precision was 97.0%, and that the recall was 89.5%, which were higher than those of mainstream models such as YOLOv5, YOLOv3, and Faster R-CNN. The results of the ablation experiments show that ASPP, new anchors, C3SAB, and C3DAB designed in this study can improve the accuracy of the model by 5.9%. Furthermore, when the ASPP dilated convolution was set to (1,5,9,13) and the loss function was set to CIoU, the model had the highest accuracy. The class activation map function was utilized to visualize the model's feature extraction results and to explain the model's region of interest for cow images in natural scenes, which demonstrates the effectiveness of the model. Therefore, the model proposed in this study can improve the accuracy of the model for detecting cow estrus events. Additionally, the model's inference speed was 71 frames per second (fps), which meets the requirements of fast and accurate detection of cow estrus events in natural scenes and all-weather conditions.

**Keywords:** cow estrus; mounting behavior detection; YOLOv5; multiscale optimization; loss-function optimization

## 1. Introduction

To meet the needs of the rapid development of dairy farming, scientific theories and sophisticated instruments are considered guaranteed for the development of dairy farming. Modern pastures that adopt intensive and large-scale breeding methods can accurately monitor the various signs and behavioral parameters of dairy cows, thereby ensuring the health of dairy cows. Automatic monitoring technology of cow estrus behavior helps improve milk production and the reproductive rate of cows. With the development of modern information and automation technology, the automatic identification of cows in estrus can remind breeders to carry out breeding in a timely manner and solve the problems of vacancy and missed detection [1]. The automatic detection technology of cow estrus can reduce the workload of ranch workers and has a very high research value [2].

The two main methods for monitoring the estrus behavior of dairy cows are a contact method based on electronic sensors, and a non-contact method based on computer vision.

With the development of electronic sensor technology and wireless network technology, accelerometers are used to detect the activity of dairy cows by examining the regularity of dairy cow estrus activity. Whether a cow is in estrus can be determined by monitoring the activity and behavioral changes in cows with pedometers and accelerometers, based on which ovulation time and optimal insemination time can be estimated [3,4]. Through verification of the rectal touch detection method, the accuracy of detecting estrus with an acceleration sensor can exceed 90% [5,6]. The acceleration sensor collar worn on the necks of dairy cows can be utilized for estrus monitoring and breeding time prediction, and the power consumption and communication effects of the neck tags were acceptable for indoor-housing conditions [7]. Temperature sensors can be installed to monitor changes in ear temperature and vaginal temperature during estrus, which helps identify the estrus status of each cow [8]. In the above studies, it is necessary that each cow wear a sensor device and to determine the cow's estrus by monitoring the changes in one or more factors, such as the cow's body temperature, exercise, and lying behavior. The research on estrus behavior detection methods of dairy cows based on electronic sensors is mainly aimed at natural breeding scenes and has good robustness. However, the higher the number of sensors on a cow, the higher the cost of the farm and the greater the stress response of the cow.

In recent years, the continuous development of smart animal husbandry, machine learning technology, video surveillance and intelligent analysis technology have helped to overcome the drawbacks of contact sensors. Video analysis technology and image technology, such as cow image segmentation [9] and cow individual detection [10], are often applied to cow estrus recognition to realize noncontact cow estrus monitoring. However, the manual selection of cow mounting features will cause feature loss, resulting in a poor generalization ability of the algorithm in dense environments.

As an autonomous feature-learning method, deep learning can avoid the limitations of manual feature extraction. Deep-learning algorithms have the advantages of fast recognition speed, high recognition accuracy, and strong model-transfer ability, so they show great potential in agricultural object classification and behavior detection [11–13]. Zhang et al. [14] proposed a deep-learning network based on bilateral filtering enhancement of thermal images and YOLOv3 [15]. The combination of image frames and optical flow has been applied to pig behavior recognition [16]. Faster R-CNN [17] and XGBoost algorithms have been utilized to identify pig mounting behavior [18]. CNNs have also been employed to identify the drinking and playing behaviors of live pigs and the behaviors of sternal lying, lateral lying, sitting, standing, and walking of sows [19,20]. In terms of cow behavior, surveillance cameras can be installed to capture all posed images of cows, which are then used to train convolutional neural networks (CNNs) to effectively identify rumination and other behaviors [21]. The model combining CNN and LSTM can also be utilized to identify the daily behaviors of cows, such as lying, standing, walking, drinking, climbing, and feeding in complex environments [22–24]. Wang et al. [25] used cow mounting behavior data to train the YOLOv3 model and to recognize cow estrus behavior

with a detection speed of 31 fps. The application of CNNs in animal husbandry research provides a reference and feasibility basis for the identification of estrus behavior in dairy cows. Therefore, it is feasible to judge whether a cow is in estrus by the cow's mounting behavior. However, current research on noncontact estrus detection of dairy cows based on deep learning only detects the mounting behavior of dairy cows with a simple background, single angle, and single size. Cow images obtained in natural breeding environments are often characterized by dense cows, complex backgrounds, and large illumination changes. At present, no researchers have studied the detection of estrus behavior of dairy cows based on deep learning in natural breeding scenarios. Existing studies are only applicable to ideal environments with few cows and have low accuracy and inference speed in natural farming scenes.

The detection of estrus behavior in dairy cows in natural breeding scenarios requires high inference speed and accuracy. The object-detection model mainly includes a one-stage method and a two-stage method. Faster R-CNN has the best inference speed and accuracy in the one-stage object detection model, and the two-stage detection model mainly includes YOLOv1-v5 [26]. Among them, the inference speed and accuracy of YOLOv5 are not only better than the YOLO series models, but also better than Faster R-CNN. Therefore, YOLOv5 was selected as the basic model in this study, as it can meet the requirement of inference speed in natural farming scenarios. To improve the model accuracy, spatial pyramid pooling (SPP) is often used to solve multi-scale problems, but when the receptive field is larger, the image feature loss is more serious. ASPP [27] uses atrous convolution to balance the contradiction between receptive field and feature loss, which can extract multiscale feature information of dairy cows. The complex background in a natural breeding environment will affect the accuracy of the model, and the attention mechanism can effectively screen important information and avoid the interference of redundant features. Most of the attention mechanisms choose to add complexity to improve performance, but sparse attention backtracking (SAB) is a new method commonly used for temporal feature extraction that incorporates a differentiable sparse attention mechanism to select from previous states [28]. In the field of image segmentation, asymmetric bottleneck modules based on depth-wise separable convolutions have been used to improve the model's feature extraction capability for contextual information but have not been introduced into the field of object detection. These optimization methods can not only optimize the inference speed of the model, but also improve the accuracy of the model, which provides ideas for improving the inference speed and accuracy of cow estrus detection.

Aimed at the problem of low accuracy and inference speed of cow estrus behavior detection in natural scenes, this paper proposes a method for cow estrus behavior detection in natural scenes based on improved YOLOv5 (CEBD-YOLO). To improve the receptive field of the model and the ability to perceive global context multiscale information, ASPP is added to the YOLOv5l network for multiscale optimization. Furthermore, the channel attention mechanism and deep asymmetric bottleneck module are applied in the estrus behavior detection model of dairy cows. Finally, CIoU and K-means are used to optimize the training process of CEBD-YOLO. This study proposes a cow estrus detection model for natural breeding scenes with multiple cows and designs several improved modules to achieve higher accuracy and inference speed, which is suitable for all-weather cow estrus detection in large farms.

## 2. Data Collection and Analysis

### 2.1. Data Sources

The cow experimental data was collected in Beijing Dadi Qunsheng Dairy Cows Breeding Base, which is located in Yanqing District, Beijing, China. The 70 m × 26 m youth cowshed was selected for installing cameras, with an installation height of 4.5 m and a field of view covering the entire youth cowshed. The camera was a high-definition infrared night vision camera (DS-2CD3T46WDV3-I3, Hikvision, Hangzhou, China), with a focal length of 6 mm, a pixel count of 4 million, a resolution of 2560 × 1440 (pixels), and a frame rate of

25 fps. The images captured during the day and at night were all RGB color images, and the captured video was stored by a hard-disk video recorder (NVR, HIKVISIONDS-8832N-K8, Hikvision, Hangzhou, China), as shown in Figure 1.

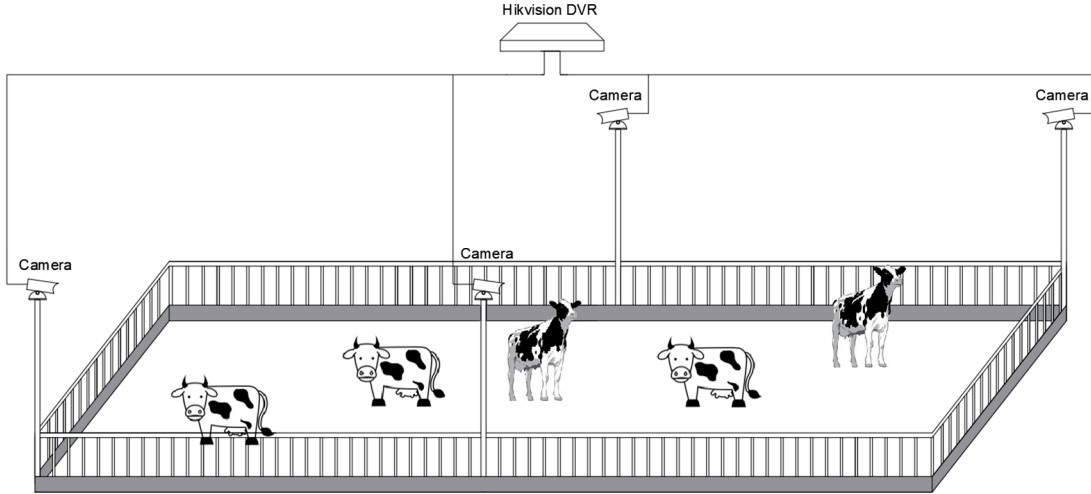

**Figure 1.** Schematic diagram of installation of estrus data acquisition equipment.

### 2.2. Data Analysis and Preprocessing

From 26 September 2021 to 9 October 2021, 24 h of video was recorded every day, and a total of 14 days of video data was collected. The cow mounting images were extracted from the cow surveillance video, and then labeling software was used to manually mark the mounting area to create a cow mounting behavior dataset. To ensure that the mounting videos of the training set and the test set came from different scenes, the dataset was divided according to the ratio of 8:2. Among them, 2668 images extracted from the first 115 mounting videos were used as the training set, and 675 mounting images extracted from the last 29 mounting videos were used as the test set, in order to more objectively test the generalization of the model in dense scenes, as shown in Table 1.

**Table 1.** Cow mounting datasets.

| Growth Periods | Number of Videos | Number of Images | Image Enhancement Methods | Image Resolution |
|---|---|---|---|---|
| Training set | 115 videos | 2668 images | Mosaic Enhancement | 2560 × 1440 |
| Test set | 29 videos | 675 images | —— | 2560 × 1440 |
| Total | 144 videos | 3343 images | —— | 2560 × 1440 |

The cow mounting behavior dataset in dense scenes contained rich lighting information. To adapt to the large changes in the scale and illumination of the cows in the dense scenes, the Mosaic method was used to enhance the cow mounting training set. The training set enhancement process is shown in Figure 2. Firstly, the original image with size of 2560 pixels × 1440 pixels was scaled to an image of size 640 pixels × 640 pixels. Four images of size 640 pixels × 640 pixels were randomly extracted and stitched into images of size 1280 pixels × 1280 pixels. The stitched image was randomly scaled and panned, and then the panned image was randomly cropped to a size of 640 pixels by 640 pixels, which is called Mosaic enhancement. To adapt to the complex environment in the natural scene, the hue (H), saturation (S), and value (V) of the images were randomly adjusted to increase the diversity of the dataset. Finally, 50% of the images were flipped vertically to complete the enhancement of the training set.

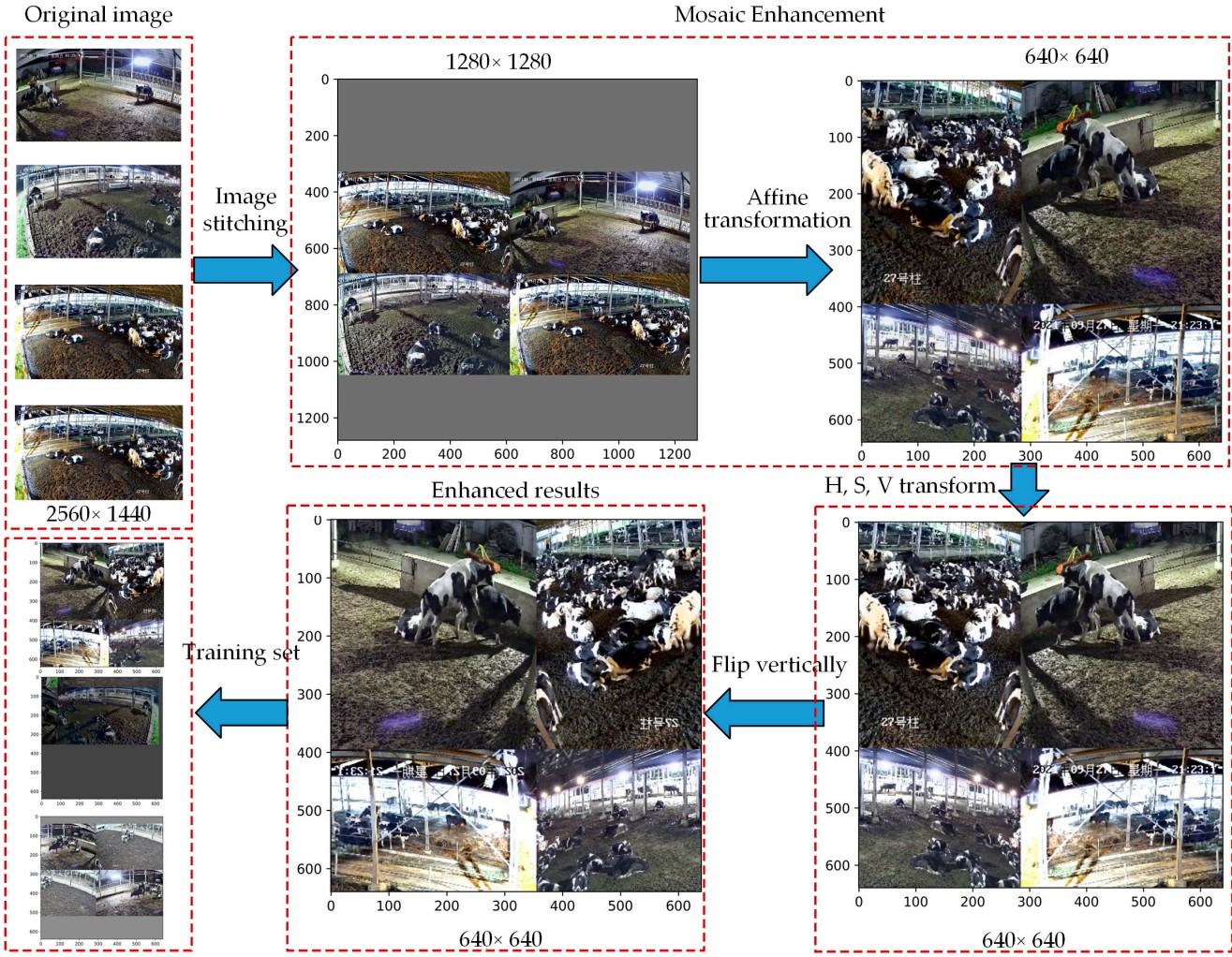

**Figure 2.** Training set enhancement process.

To explore the characteristics of cow climbing images captured by fixed cameras in dense scenes, a mounting and non-mounting image were selected from each scene to form an image pair, and a total of 162 pairs of images were obtained. Structural Similarity (SSIM) and Mean Squared Error (MSE) were used to calculate the similarity between each pair of images and plot the similarity between each pair of images. The distribution of SSIM and MSE values between each pair of images is shown in Figure 3a,b. Image pairs with SSIM values greater than 0.8 accounted for 90.74%, and image pairs with MSE values less than 600 accounted for 90.12%, indicating that the images of the mounting cows and non-mounting cows in the same scene had high structural similarity and pixel value similarity. According to the similarity statistics shown in Figure 3d, when the mounting behavior occurred, the global features of the images changed little, and the proportion of similar image pairs was higher than 90%. Figure 3c shows the visualization results of the climbing and non-climbing image pairs in the same scene, where the SSIM value was 0.99 and the MSE value was 70.03, which further proves that the image pair was highly similar, and it was difficult for human eyes to distinguish. The analysis results show that due to the large shooting range of the camera, when the cow mounting event occurred in the same dense scene, the change in the global image features was small, and the model needed to be strengthened to extract the detailed information of the image.

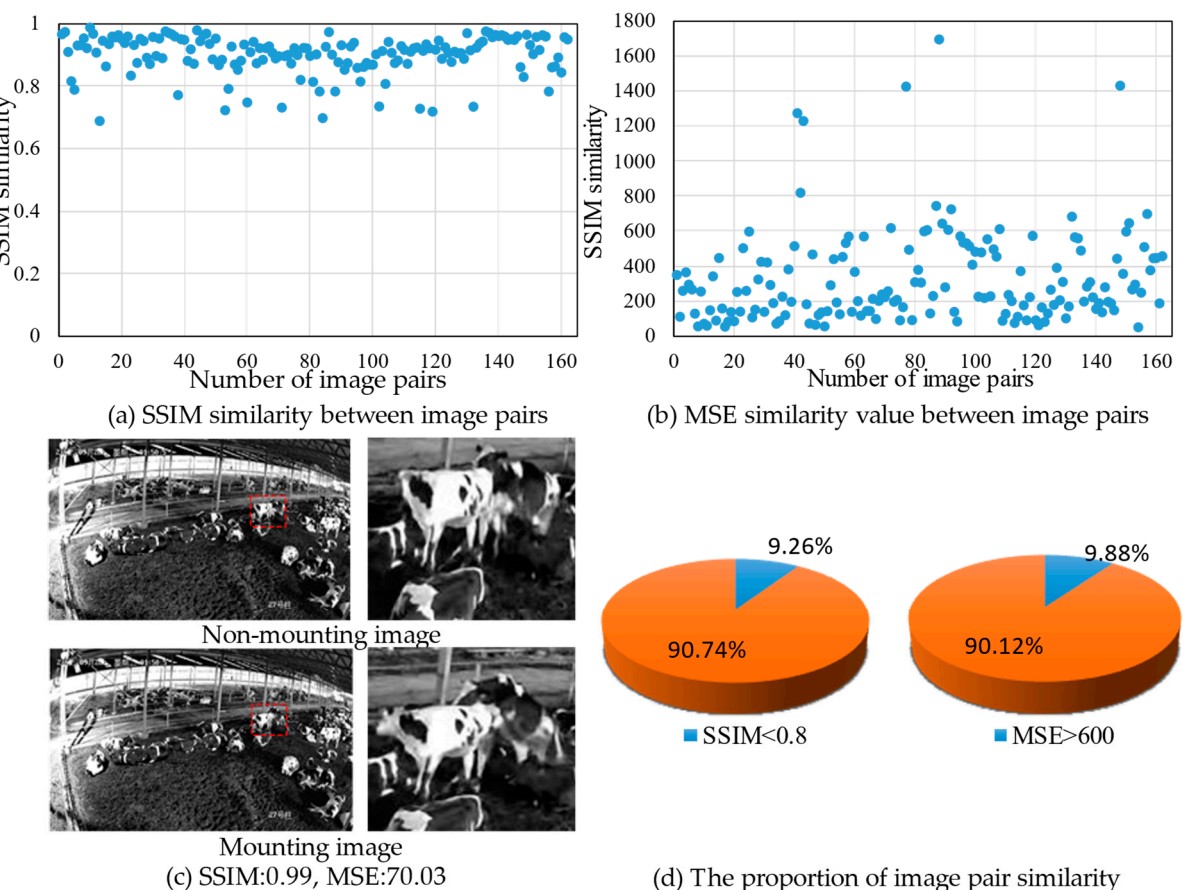

Figure 3. Similarity calculation results between image pairs.

## 3. Methods

Currently, YOLOv5 is the object detection network with the best performance, and the detection speed and detection accuracy are greatly improved compared with other models. The network structure of YOLOv5 is mainly composed of three parts: a feature extraction backbone module, a neck network, and a prediction module. The overall structure is shown in Figure 4.

The feature extraction backbone module of YOLOv5 is mainly composed of cross-stage partial (CSP) modules. The CSP module is composed of residual units and convolution units and the structure is shown in Figure 4. The backbone network is formed by stacking different numbers of CSPs, which is mainly used for feature extraction in the model. The focus module consists of a slice layer and a convolution layer. The focus module slices RGB images with an input of $640 \times 640 \times 3$ by interval sampling and obtains four feature images with a size of $320 \times 320 \times 3$. These four feature maps are then spliced into $320 \times 320 \times 12$ feature maps in the channel dimension, which transforms the H-W spatial information into a channel space. Next, the shallow detail information of the image in the multidimensional channel is extracted through the convolution layer, and the feature map with a size of $320 \times 320 \times 16$ is outputted. This focus module can effectively reduce the loss of image information in the process of downsampling, reduce the amount of calculation, and improve the calculation speed. Therefore, using focus as the initial layer of the feature extraction backbone can enlarge the receptive field and retain more of the original information.

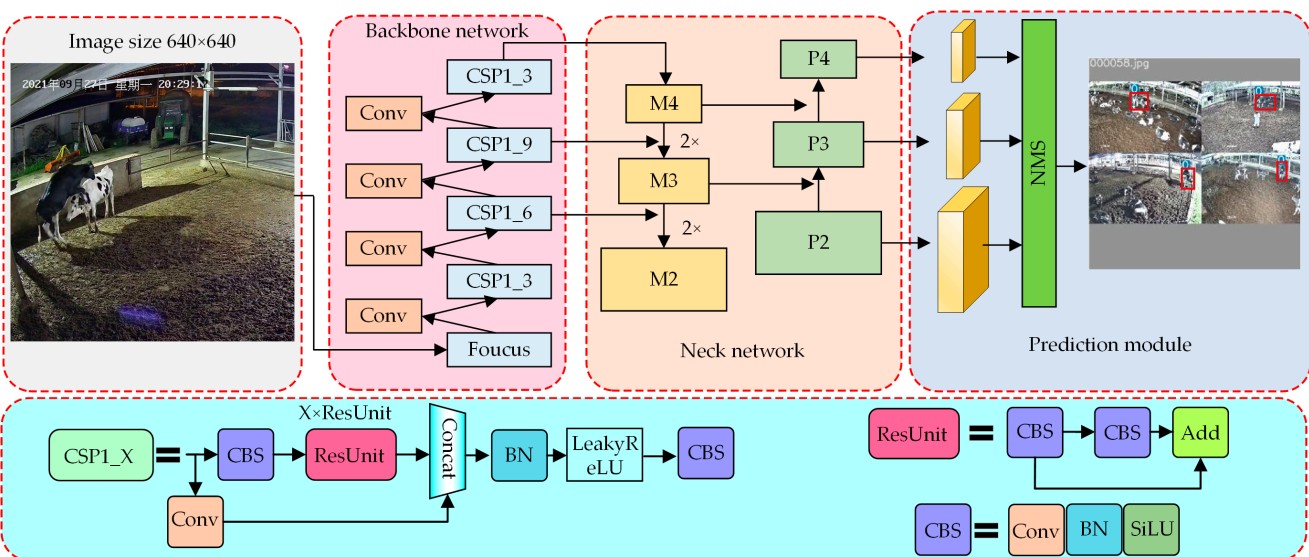

**Figure 4.** Network structure of YOLOv5.

The neck is a cyclic pyramid structure composed of a convolution operation, an upsampling operation, and CSP, which can fuse different feature layers of an image to make mask predictions. The neck consists of feature pyramid networks (FPN) and path aggregation networks (PAN). FPN contains feature maps of three scales, from M2 to M4, and uses a top-down approach to fuse different levels of semantic information extracted by the backbone network. PAN contains feature maps of three scales, from P2 to P4, and uses a bottom-up approach to fuse semantic information from low-level to high-level. The neck network can simultaneously fuse shallow and deep feature information to effectively improve the performance of the detector. The prediction box is obtained through non-maximum suppression (NMS).

Affected by factors such as occlusion, shooting angle, and complex background, the appearance information of mounting cows is insufficient, and the scale of mounting cows varies. ASPP increases the receptive field by introducing a new parameter dilation rate, which realizes feature sampling with different intervals and can better learn the global and local information of cow mounting images. The calculation methods of the atrous convolution kernel and the size of the receptive field are shown in Formulas (1) and (2).

$$f'_k = f_k + (f_k - 1) * (Rate - 1) \tag{1}$$

$$R_m = R_{m+1} + (f'_k - 1) * \prod_{i=1}^{m-1} s_i \tag{2}$$

where $f_k$ represents the original convolution kernel size, $f'_k$ represents the atrous convolution kernel size, *Rate* represents the dilation rate of the convolution kernel, $R_m$ represents the *m*-th layer receptive field size after the atrous convolution, $s_i$ represents the stride of the *i*-th layer, and '$*$' represents the convolution operation.

The ASPP-net extracts characteristic graphs of different sensory field sizes by pooling layers of different sizes to combine global and subregional information. The number of channels in the feature graph is widened to provide effective global context information. To save more detailed information and adapt to the multiscale cows' mounting behavior, five ASPP modules with different atrous convolution combinations are designed in this paper. The combinations of atrous convolutions (1,2,3,4), (1,3,5,7), (1,5,9,13), (1,6,12,18), and (6,12,18,24) are analyzed to explore the most suitable combination of atrous convolutions for multiscale cow mounting behavior, which makes the model more robust to deformed cows. The structure of the ASPP module is shown in Figure 5.

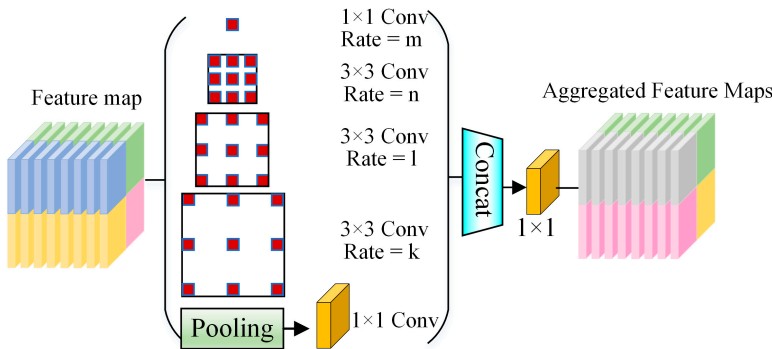

(m,n,l,k): (1,2,3,4), (1,3,5,7), (1,5,9,13), (1,6,12,18), (6,12,18,24)

**Figure 5.** Structure of the ASPP module.

SAB makes changes to traditional networks. During forward inference, the system will manage a memory unit and select at most a sparse subset of past memory, which is referred to as sparse retrieval. During backpropagation, gradients are propagated through memory and a sparse subset of surrounding cells (referred to as sparse replay). Collectively, this finding suggests that better generalization performance can be produced with improved memory and long-term dependencies.

In this paper, SAB was introduced to target detection, which extracts sparse original attention from the input feature map. The original attention values are then sorted from large to small, and the sparse attention values are normalized by subtracting the attention values that rank first $k_{top} + 1$. The normalized result is input into the ReLU function to obtain the sparse attention value. As previously mentioned, when SAB is utilized in image feature extraction, global dependencies can be successfully captured.

Inspired by the depth-wise asymmetric bottleneck (DAB) [29] in the field of semantic segmentation, this paper reconstructs the DAB by using this framework, asymmetric convolution and deep separable convolution. The DAB structure is shown in Figure 6.

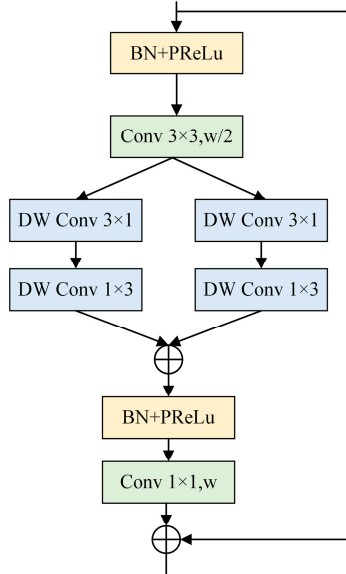

**Figure 6.** Structure diagram of the DAB module.

A deep network can increase the receptive field and extract more complex features, but it will also be accompanied by the problem of gradient explosion and gradient disappearance. Therefore, this paper uses two branches to extract module features in parallel. By stacking $1 \times 3$ and $3 \times 1$ asymmetric depth-wise separable convolutions (DW Conv),

the model receptive field is improved, and local and contextual information is extracted, thereby reducing model parameters and redundant feature extraction. In the model, the batch normalization (BN) operation and parametric rectified linear unit (*PReLu*) activation function are employed to extract nonlinear features of the feature map to avoid gradient disappearance and gradient explosion. The calculation formula of *PReLu* is shown in Formula (3).

$$PReLu(x_i) = \begin{cases} x_i & if \ x_i > 0 \\ a_i x_i & if \ x_i \leq 0 \end{cases} \tag{3}$$

where *i* denotes different channels and *x* denotes the feature map of the input.

At the end of the DAB module, $1 \times 1$ standard convolution is used to restore the feature map size, so the number of convolution kernels is set to *w*. In summary, DAB not only improves the receptive field of the model but also extracts complex features from deep features. The module is very lightweight.

The data analysis results presented in Section 2.1 show that in the dairy cow group breeding environment, the scale of mounting cows was small, and the cow mounting characteristics were not obvious. If too much attention is given to the global features of the image, the details of the image will be lost. In contrast, if too much attention is given to the details, the changes in the global features of the image cannot be recorded. Adding the ASPP module to the YOLOv5 model not only improves the feature extraction capability of the model but also expands the model's receptive field and retains the detailed features of the image. The C3SAB_3 module is constructed by combining the attention mechanism SAB and the C3_3 module, and the C3SAB_3 module is connected with ASPP to compensate for the feature loss when the receptive field of the model is increased by the atrous convolution in ASPP. The C3DAB_3 module is constructed by combining the DAB and C3_3 modules and is applied to the high-level semantic feature extraction of the model, which can fully extract the model context information without affecting the model parameters and inference speed. In summary, using the YOLOv5, ASPP, C3SAB_3 and C3DAB_3 modules, this paper proposes an improved YOLOv5 model for cow estrus behavior detection in natural scenes (CEBD-YOLO) to improve the accuracy of cow mounting event detection. The model structure parameters are shown in Table 2, and the model structure is shown in Figure 7.

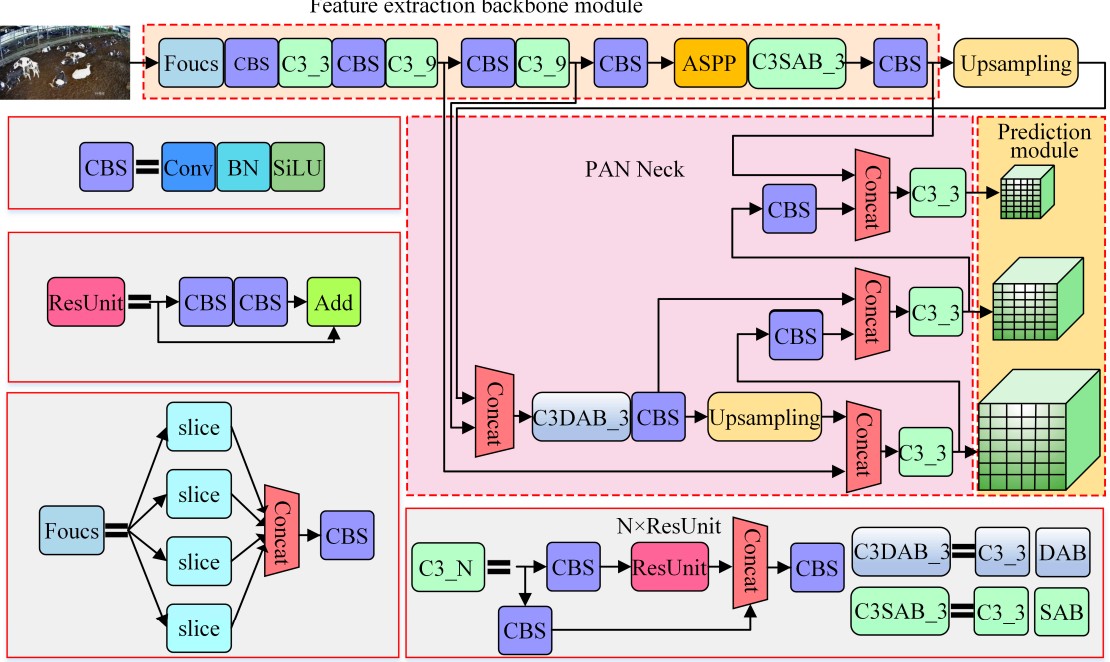

**Figure 7.** Overall structure of the CEBD-YOLO model.

**Table 2.** Network parameters.

| Layers | Network Layer | Input Size | Step | Number of Channels |
|--------|---------------|------------|------|--------------------|
| 1 | Focus | $640 \times 640 \times 3$ | —— | 64 |
| 2 | CBS $3 \times 3$ | $320 \times 320 \times 64$ | 2 | 128 |
| 3 | C3_3 | $160 \times 160 \times 128$ | —— | 128 |
| 4 | CBS $3 \times 3$ | $160 \times 160 \times 128$ | 2 | 256 |
| 5 | C3_9 | $80 \times 80 \times 256$ | —— | 256 |
| 6 | CBS $3 \times 3$ | $80 \times 80 \times 256$ | 2 | 512 |
| 7 | C3_9 | $40 \times 40 \times 512$ | —— | 512 |
| 8 | CBS $3 \times 3$ | $40 \times 40 \times 512$ | 2 | 1024 |
| 9 | ASPP | $20 \times 20 \times 1024$ | —— | 1024 |
| 10 | C3SAB_3 | $20 \times 20 \times 1024$ | —— | 1024 |
| 11 | CBS $1 \times 1$ | $20 \times 20 \times 1024$ | 1 | 512 |
| 12 | Upsample | $20 \times 20 \times 512$ | —— | —— |
| 13 | Concat | —— | —— | —— |
| 14 | C3DAB_3 | $40 \times 40 \times 512$ | | 512 |
| 15 | CBS $1 \times 1$ | $40 \times 40 \times 512$ | 1 | 256 |
| 16 | Upsample | $40 \times 40 \times 256$ | —— | |
| 17 | Concat | —— | —— | |
| 18 | C3_3 | $80 \times 80 \times 256$ | | 256 |
| 19 | CBS $3 \times 3$ | $80 \times 80 \times 256$ | 2 | 256 |
| 20 | Concat | —— | —— | —— |
| 21 | C3_3 | $80 \times 80 \times 512$ | —— | 512 |
| 22 | CBS $3 \times 3$ | $80 \times 80 \times 512$ | 2 | 512 |
| 23 | Concat | —— | —— | —— |
| 24 | C3_3 | $80 \times 80 \times 512$ | —— | 1024 |
| 25 | Detect | —— | —— | —— |

The overall structure of the CEBD-YOLO model consists of three parts: the feature extraction backbone module, Pyramid Attention Network Neck (PAN Neck), and prediction module. First, after the cow mounting image is input into the CEBD-YOLO model, the image features are extracted through the feature extraction backbone module. Second, PAN Neck is selected to achieve different levels of semantic feature aggregation to perceive the cow mounting on different scales. Last, the multiscale aggregated features are input into the prediction module to detect the mounting behavior of cows.

The feature extraction backbone module proposed in this study is mainly composed of the Focus, CBS, and C3 modules. The feature extraction backbone is constructed by stacking C3 and CBS to extract the features from the cow mounting images.

As an important part of the feature extraction backbone, CBS includes three parts: convolution operation, normalization operation, and *SiLU* activation function. In cow herd images collected in natural scenes, the distribution of cows is relatively dense. Therefore, this paper adopts a smooth and nonmonotonic *SiLU* function that can more fully extract image details. Assuming that the input of the previous layer is $x$, the *SiLU* activation function can be expressed as Formula (4):

$$SiLU(x) = x \cdot Sigmoid(x) \tag{4}$$

$$Sigmoid(x) = \frac{1}{1 + \exp(-x)} \tag{5}$$

As shown in Figure 7, C3 is mainly composed of CBS and ResUnit, and C3_N indicates that the C3 module is composed of N ResUnits. To reduce feature loss, skip connections are used multiple times in the C3 module to fuse semantic information at different levels.

In this study, the anchors of YOLOv5 were obtained according to the clustering on the COCO dataset, and CEBD-YOLO was developed to identify mounting behavior in natural farming scenes. K-means clustering is the most well-known clustering algorithm with simplicity and efficiency and is the most widely used of all clustering algorithms. Given a

set of data points and the required number of clusters k, where k is specified by the user, the K-means algorithm repeatedly divides the data into k clusters according to a certain distance function. Therefore, to make the model more suitable for cow estrus detection, the K-means algorithm is applied to generate new anchors. The original anchor sizes are $(10 \times 13)$, $(16 \times 30)$, $(33 \times 23)$, $(30 \times 61)$, $(62 \times 45)$, $(59 \times 119)$, $(116 \times 90)$, $(156 \times 198)$, and $(373 \times 326)$. The original anchor sizes are modified to $(30 \times 39)$, $(56 \times 49)$, $(48 \times 72)$, $(65 \times 85)$, $(102 \times 72)$ $(103 \times 101)$, $(115 \times 180)$, $(157 \times 135)$, and $(215 \times 143)$.

In terms of quantitative evaluation, precision (*P*), recall (*R*), and mean average precision (*mAP*) were selected as the evaluation indicators. The range of precision and recall is [0, 1], and the calculation was performed using Formulas (6) and (7).

$$R = TP_1 / (TP_1 + FN_1) \tag{6}$$

$$P = TP_1 / (TP_1 + FP_1) \tag{7}$$

$TP_1$, $FP_1$, and $FN_1$ are the number of true positives, false positives, and false negatives, respectively. Average precision (*AP*) represents the average precision of each class. The calculation formula is expressed as Formula (8).

$$AP = \int_0^1 P(r)dr \tag{8}$$

*M* is the number of classes, and *mAP* is the average of all classified APs. The higher the value of *mAP*, the better the detection capability of cows' mounting behavior. The calculation is performed using Formula (9).

$$mAP = \frac{\sum_{c=1}^{M} AP(c)}{M} \tag{9}$$

## 4. Experimental Results and Analysis

### 4.1. Experimental Parameter Settings

A 16-GB NVIDIA Tesla P100 GPU was used for training, and a deep learning algorithm training platform was built based on the Ubuntu 16.0 operating system, Python 3.8, and PyTorch 1.7.1. The CUDA API version 10.1 and the CuDNN version 8.0.5 were used. The initial learning rate in the training process was set as 0.01, and the cosine annealing strategy was used to reduce the learning rate. The input image size, batch size, and the number of epochs were set as $640 \times 640$, 40, and 200, respectively.

### 4.2. Base Model Selection

Four models, namely YOLOv5s, YOLOv5m, YOLOv5l, and YOLOv5x, having different computation complexities and numbers of parameters, were constructed using basic backbone networks with different widths and depths. To select the base model suitable for estrus behavior detection in dairy cows, the transfer learning method was employed for using the pretrained weights obtained in the COCO dataset as the initial weights to learn the mounting behavior of dairy cows, and then the test set was used to analyze the four models. The test results are presented in Table 3.

**Table 3.** Detection results of different YOLOv5 models on the cows' mounting behavior.

| Model | Precision/% | Recall/% | mAP/% (IoU = 0.5:0.95) | mAP/% (IoU = 0.5) | Speed/ms | Weight/MB | Parameter/M | GFLOPS |
|---|---|---|---|---|---|---|---|---|
| YOLOv5s | 86.5 | 83.6 | 36.7 | 85.6 | 4.4 | 13.7 | 7.2 | 16.5 |
| YOLOv5m | 91.5 | 81.5 | 42.3 | 86.3 | 7.9 | 40.4 | 21.2 | 49.0 |
| YOLOv5l | 91.9 | 87.7 | 49.6 | 88.4 | 12.9 | 89.3 | 46.5 | 109.1 |
| YOLOv5x | 92.1 | 82.8 | 45.3 | 89.5 | 20 | 166.9 | 86.7 | 205.7 |

The training set was used to train different YOLOv5 models. After each epoch of training, the model was tested on the test set, and the model with the best test results was selected. The best results for each YOLOv5 model are listed in Table 3. When the IoU is 0.5, the mAP is used to represent the accuracy of the model and the giga floating-point operations per second (GFLOPS) is used to represent the computational load of the model. YOLOv5s exhibited the smallest model weights and number of parameters, and YOLOv5x had the largest model weights and number of parameters and the highest mAP (IoU = 0.5) of 89.5%, thus indicating that with an increase in the depth and width of the model, the accuracy of the detection of cow mounting behavior improves. YOLOv5l exhibited the highest mAP (IoU = 0.5:0.95), indicating its strong generalization ability; moreover, the detection speed was 1.5 times faster than that of YOLOv5x, but the model weight size was 89.3 Mbyte (MB) and the parameter size was 46.5 M, which were lighter than YOLOv5x. Considering the need to balance model accuracy, detection speed, and hardware resource consumption for the detection of cow mounting behavior, in this study, YOLOv5l was selected as the base model and was then improved to be suitable for cow mounting behavior detection.

*4.3. Model Training Results*

The CEBD-YOLO model proposed in this paper was obtained by improving the YOLOv5l model, and the CEBD-YOLO model was trained using the cow mounting behavior training set. After each epoch of training, the test set was used to test the model detection effect. The changes in the value of the loss function during the training process is shown in Figure 8.

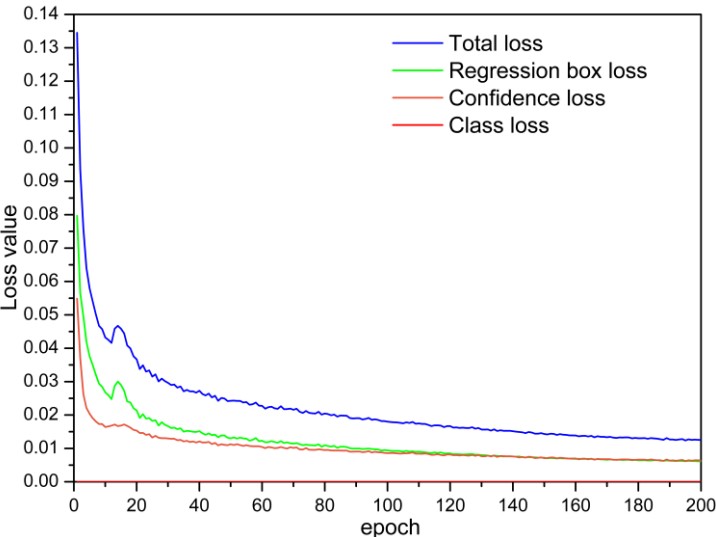

**Figure 8.** Change trends of the loss functions during the training process.

With an increase in the number of training epochs, all loss functions exhibited a downward trend and then gradually became stable (Figure 8). The confidence loss and regression box loss converged to approximately 0.01, and the total loss finally converged to approximately 0.02. In summary, after 200 rounds of training, the CEBD-YOLO model converged to an ideal degree.

After each round of training, the model was tested on the test set, and the accuracy of the model was verified using four evaluation indicators: precision, recall, mAP (IoU = 0.5:0.95), and mAP (IoU = 0.5). The changes in the values of the evaluation indicators during the training process are shown in Figure 9. All four evaluation indicators initially increased, then oscillated within a certain range, and finally became stable. mAP (IoU = 0.5:0.95) was stable at approximately 0.5, and the other three indicators were stable in the range of 0.8–1.0, which proves that the model yielded high accuracy and good

convergence in the detection of cow mounting behavior in the test set. The model with the highest mAP (IoU = 0.5) during training was selected as the best result of CEBD-YOLO.

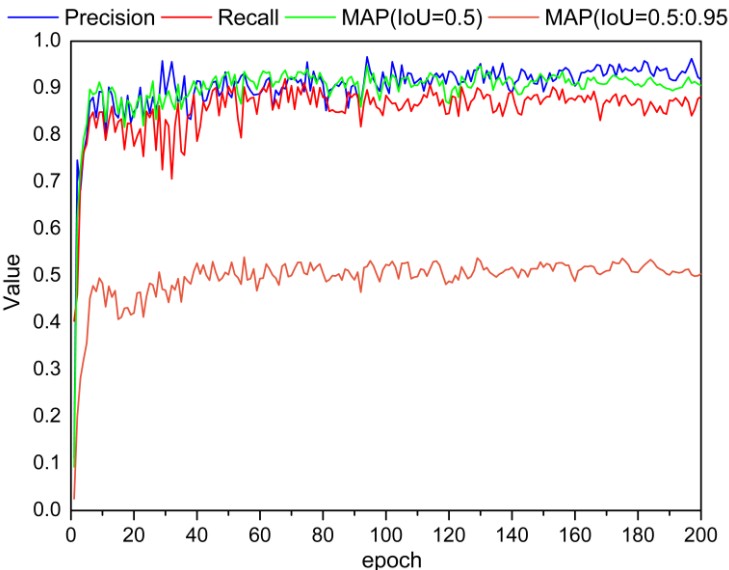

**Figure 9.** Changes in the values of the evaluation indicators of the model in the test set during the training process.

*4.4. Comparison of Detection Results of Different Models*

Presently, the more popular target detection networks include Faster R-CNN, YOLOv3, and YOLOv5. After these models were retrained by the cow mounting behavior training set, the detection effects of the models were tested in the test set. Parameters such as mAP, inference speed, and weight size were used to evaluate the CEBD-YOLO model and other models. The comparison results are shown in Table 4.

**Table 4.** Cow mounting behavior detection results obtained using different models.

| Model | mAP (IoU = 0.5:0.95)/% | mAP (IoU = 0.5)/% | Speed/ms | Weight/MB |
|---|---|---|---|---|
| Faster RCNN | 46.2 | 83.6 | 91.7 | 315.0 |
| YOLOv3 | 36.5 | 83.9 | 14.3 | 117.67 |
| YOLOv5l | 49.6 | 88.4 | 12.9 | 89.3 |
| Ours (CEBD-YOLO) | 51.9 | 94.3 | 14.1 | 154.9 |

The mAP (IoU = 0.5) value of CEBD-YOLO was 5.9% higher than that of YOLOv5l and 10.4% and 10.7% higher than that of YOLOv3 and Faster R-CNN, respectively. The mAP (IoU = 0.5:0.95) value of CEBD-YOLO was the highest, which proves that the model proposed in this paper had better generalization and detection effects. Although the size of the improved model was 154.9 MB, the inference time of each image was only 14.1 ms, and approximately 71 images were detected per second, which is approximately double the inference speed indicated in the literature [24]. The real-time requirements of cow estrus behavior detection were met.

The detection results of different models, which show different detection results for different scales of cow mounting images, are shown in Figure 10. Each model had a better detection effect on large objects, while the YOLO series models had a higher missed-detection rate for small and medium objects, and the Faster R-CNN had a higher false-detection rate. The CEBD-YOLO model proposed in this paper had the best detection effect on large, medium, and small targets, and the overall detection accuracy in the test set reached 93.4%. Compared with other models, CEBD-YOLO had fewer missed detections

and false detections, which verifies the effectiveness of the improved method proposed in this paper.

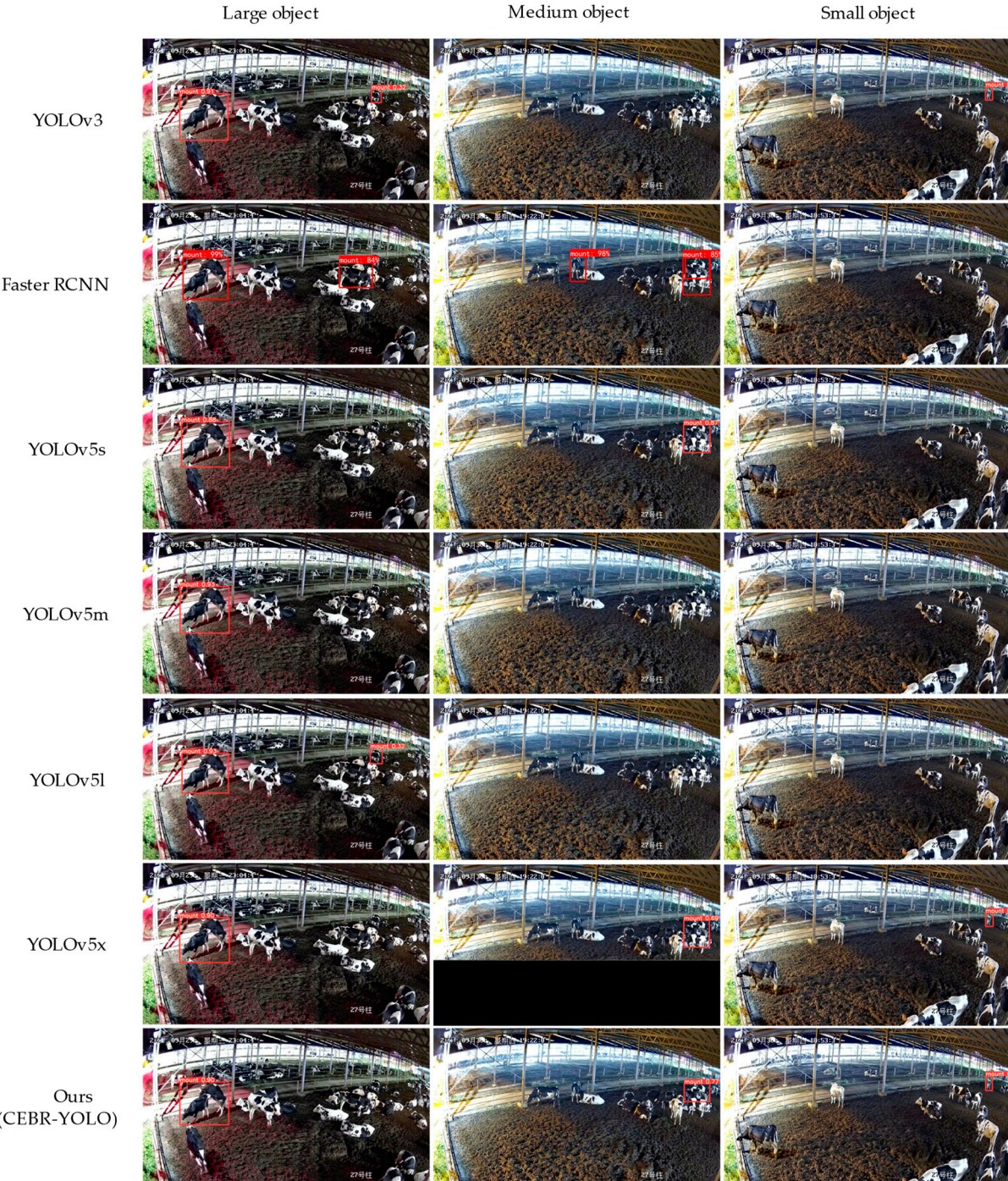

**Figure 10.** Visualization of the test results of different models.

### 4.5. Ablation Experiment

### 4.5.1. Comparison of the Ablation Results of Different Optimization Modules

The CEBD-YOLO model was constructed from YOLOv5l, the new anchors, ASPP, and the C3SAB_3 and C3DAB_3 modules. Therefore, to verify the improvement effect of each improved module on the model, the control variable method was selected to design the ablation comparison experiments. The training set was used to train the model, and the detection results of mounting behavior were tested in the test set.

The experimental results of the ablation comparison of different optimization modules are shown in Table 5. When new anchors were added to the YOLOv5l model, the value of mAP (IoU = 0.5) increased by 2.8%. This value increased by 5.0% after new anchors and the ASPP optimization module were added. When the C3SAB module was added to the model, the accuracy of YOLOv5l + New anchors + ASPP + C3SAB was 93.8%, the value of mAP (IoU = 0.5) increased by 0.4%, and the weight of the model did not change. YOLOv5l + New anchors + ASPP + C3SAB + C3DAB is the CEBD-YOLO model proposed in this paper; the value of mAP (IoU = 0.5) was 94.3%. The C3DAB module improved the value of mAP (IoU = 0.5) in the model by 0.5%. The experimental results show that each optimization module could improve the accuracy of the model and that the accuracy of CEBD-YOLO was 5.9% higher than that of the model before the improvement.

**Table 5.** The experimental results of different optimization modules' ablation comparisons.

| Number | Model | Precision/% | Recall/% | mAP/% (IoU = 0.5:0.95) | mAP/% (IoU = 0.5) | Speed/ms | Weight/MB |
|---|---|---|---|---|---|---|---|
| 1 | YOLOv5l | 91.9 | 87.7 | 49.6 | 88.4 | 12.9 | 89.3 |
| 2 | YOLOv5l + New anchors | 91.2 | 85.3 | 46.8 | 91.2 | 12.8 | 89.4 |
| 3 | YOLOv5l + New anchors + ASPP | 91.4 | 91.7 | 54.9 | 93.4 | 16.5 | 152.4 |
| 4 | YOLOv5l + New anchors + ASPP + C3SAB | 94.4 | 89.0 | 55.2 | 93.8 | 16.3 | 152.4 |
| 5 | CEBD-YOLO (YOLOv5l + New anchors + ASPP + C3SAB + C3DAB) | 97.0 | 89.5 | 51.9 | 94.3 | 14.1 | 154.9 |

### 4.5.2. Ablation Experiments with Different ASPP Modules

To obtain the best ASPP module structure, this subsection discusses the dilation rates of atrous convolutions in different branches of the ASPP module. First, we set up 5 groups of expansion rate control experiments; the expansion rate combinations were (1,2,3,4), (1,3,5,7), (1,5,9,13), (1,6,12,18), and (6,12,18,24). Table 6 shows the combined results of different expansion ratio values in ASPP. When the inflation rate was (1,5,9,13), the mAP (IoU = 0.5) value of the model for detecting the climbing behavior of cows was 94.3%. The accuracy of the model decreased if the dilation rate was too large or too small. When the dilation rate combination was (6,12,18,24), the model inference speed decreased, and the weight size increased. Thus, the best dilation rate combination in the ASPP module was (1,5,9,13).

**Table 6.** Combination results of different expansion ratio values in ASPP.

| Model | ASPP | Precision/% | Recall/% | mAP/% (IoU = 0.5:0.95) | mAP/% (IoU = 0.5) | Speed/ms | Weight/MB |
|---|---|---|---|---|---|---|---|
| CEBD-YOLO | (1,2,3,4) | 89.6 | 86.5 | 51.8 | 91.5 | 14.1 | 154.9 |
| | (1,3,5,7) | 87.2 | 88.6 | 49.7 | 91.9 | 14.0 | 154.9 |
| | (1,5,9,13) | 97.0 | 89.5 | 51.9 | 94.3 | 14.1 | 154.9 |
| | (1,6,12,18) | 89.2 | 90.7 | 53.1 | 92.6 | 14.1 | 154.9 |
| | (6,12,18,24) | 94.2 | 86.4 | 51.8 | 92.6 | 14.8 | 170.9 |

### 4.5.3. Ablation Experiments with Different Loss Functions

A comparison of the cow detection accuracy using the current mainstream loss functions DIoU, GIoU, alpha-IoU, and CIoU as the loss functions of the CEBD-YOLO model is shown in Table 7. For the CIoU loss function, the model had the highest accuracy, the mAP (IoU = 0.5) value was 94.3%, and different loss functions hardly affected the model inference speed and weight. These results are attributed to the large variation range of the detection frame when a cow mounts and the notion that the CIoU loss function considers the proportional relationship between two rectangular frames, which can better adapt to cow mounting behavior detection.

**Table 7.** Ablation experiment results of CEBD-YOLO by using different loss functions.

| Model | Precision/% | Recall/% | mAP/%<br>(IoU = 0.5:0.95) | mAP/%<br>(IoU = 0.5) | Speed/ms | Weight/MB |
|---|---|---|---|---|---|---|
| DIoU | 95.2 | 90.1 | 50.3 | 92.5 | 13.9 | 154.9 |
| GIoU | 93.3 | 91.4 | 52.6 | 94.1 | 14.0 | 154.9 |
| Alpha-IoU [30] | 93.3 | 88.2 | 51.3 | 91.4 | 13.9 | 154.9 |
| Ours(CIoU) | 97.0 | 89.5 | 51.9 | 94.3 | 14.1 | 154.9 |

*4.6. Discussion*

The comparative results of this study and other studies on livestock behavior detection are shown in Table 8. In this study, we collected data from a dense scene containing 200 dairy cows, and the scale of the dairy cows in the images was so large that the identification difficulty was low. However, other studies on livestock behavior detection only had a maximum of 27 dairy cows per image. When the number of livestock is small, the accuracy of the references [31–36] is lower than that of the CEBD-YOLO model, and as the number of livestock increases, the accuracy of these methods will decrease. The object of this study was a natural breeding environment containing 200 cows. The CEBD-YOLO model achieved a high accuracy of 94.3%. The previous studies shown in Table 8 used the YOLOv3, Mask R-CNN, SSD, and Faster R-CNN networks as basic models to detect livestock individuals, while the inference speed and detection accuracy of YOLOv5 chosen in this study are much higher than those networks. Table 4 shows the accuracy and inference speed of these models in the dairy cow estrus detection dataset. The inference speed and detection accuracy of the CEBD-YOLO model were much higher than those of YOLOv3 and Faster R-CNN. To sum up, the CEBD-YOLO model not only realized the automatic detection of estrus behavior of dairy cows in natural breeding scenarios with a large number of dairy cows, but also its accuracy and inference speed were much higher than previous studies on livestock behavior detection. It is proved that the improved method proposed in this study can effectively improve the detection accuracy and inference speed, and the CEBD-YOLO model can be applied to all-weather non-contact monitoring of cow estrus behavior in natural breeding scenes.

**Table 8.** Comparison of different livestock behavior detection methods.

| Studies | Year | Species | Research Contents | Objects | Method | Accuracy |
|---|---|---|---|---|---|---|
| Li et al. [31] | 2021 | goat | Multi-behavior recognition | A dairy goat | AlexNet + ResNet50 + Vgg16 | 85.6% |
| Fuentes et al. [32] | 2020 | cow | Multi-behavior recognition | 27 cows | YOLOv3 + I3D | 85.6% |
| Li et al. [33] | 2019 | pig | Mounting detection | 4 pigs | Mask R-CNN | 91.47% |
| Zhang et al. [34] | 2019 | pig | Mounting detection | 4 pigs | SSD + MobileNet | 92.3% |
| Yang et al. [18] | 2021 | pig | Mounting detection | 4 pigs | Faster R-CNN + XGBoost | 95.15% |
| Guo et al. [35] | 2019 | cow | Mounting detection | 10 cows | Computer vision | 90.9% |
| Fuentes et al. [32] | 2020 | cow | Mounting detection | 27 cows | YOLOv3 + I3D | 82.1% |
| Li et al. [36] | 2022 | cow | Multi-behavior recognition | A cow | MiCT | 91.8% |
| Ours (CEBD-YOLO) | 2022 | cow | Mounting detection | 200 cows | YOLOv5 | 94.3% |

To further analyze the detection effect of the CEBD-YOLO model proposed in this paper on the estrus behavior of dairy cows in different environments, the detection effects of the model at different scales during both day and night were tested. As the model uses three branches to detect cows of different scales and each branch depends on the feature extraction backbone module, this paper uses class activation mapping (CAM) [37] to visually analyze the feature extraction results of the last convolution layer in the feature

extraction backbone module. The visual analysis results are shown in Figure 11. The darker the red part is, the more attention the model pays to this part of the image, followed by the yellow part. The bluer the heatmap is, which means that the model considers this part redundant information, the lesser the influence on the detection of cow mounting behavior.

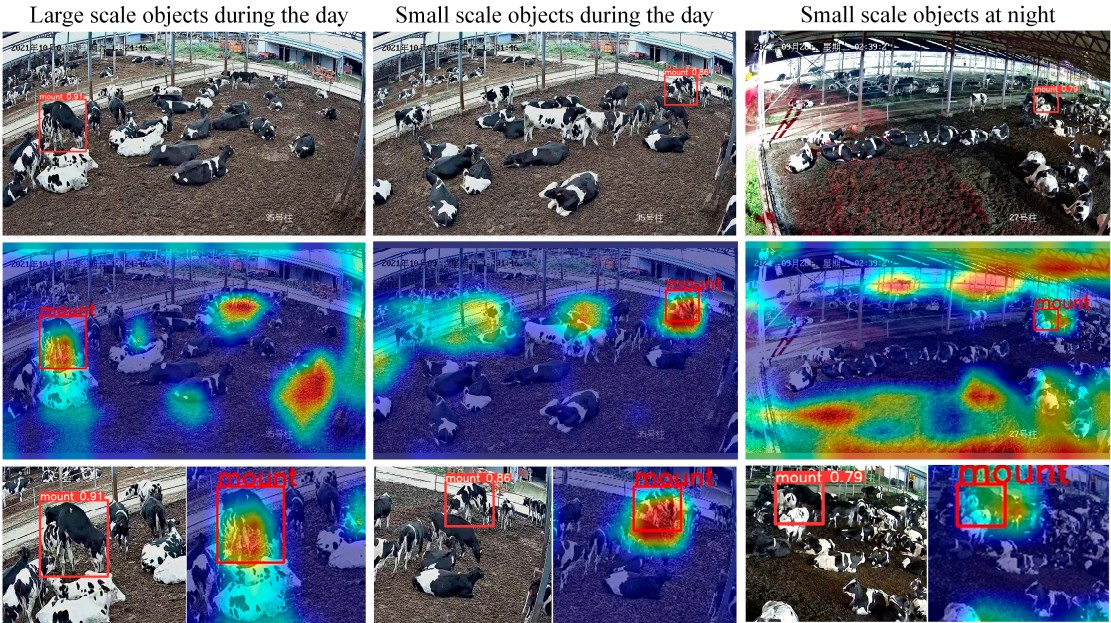

**Figure 11.** Visualization analysis of cow mounting behavior extracted by CEBD-YOLO.

According to an analysis of the model feature extraction results at different scales, the lighting conditions were better during the day, the features extracted by the model focused on the cows that were mounting, and the ground color and other cows had minimal influence on the model. At night, the scene was mainly illuminated by fluorescent lamps, the ground was dark, the model was greatly affected by the background, and less attention was given to the cows and the cows that were mounting. In addition, cows easily gather at night, so there are many small-scale cows in natural scenes, which hinders cow estrus detection. However, despite the interference of the background, lighting and other conditions, the detection effect of the model in this paper was still good for small objects in the distance and could accurately focus on the cows that were mounting, which verifies the effectiveness of the model.

## 5. Conclusions

Aimed at the problems of complex backgrounds and low accuracy and inference speed in the process of cow estrus behavior detection in natural breeding environments, this paper proposes a cow estrus behavior detection model based on an improved YOLOv5 model (CEBD-YOLO) in natural scenes. Compared with the YOLOv5x, YOLOv5l, YOLOv5m, YOLOv5s, YOLOv3, and Faster R-CNN algorithms, the CEBD-YOLO model had better performance in terms of precision, recall, and mAP. The mAP (IoU = 0.5) value was 94.3%, which is 5.9% higher than that of YOLOv5l. The precision of CEBD-YOLO was 97.0% and the recall of CEBD-YOLO was 89.5%, which is higher than that of YOLOv5l. The inference time of CEBD-YOLO was 14.1 ms per image, and the inference speed reached 71 fps, which meets the real-time performance of cow estrus detection. To adapt to the multiscale problem of estrus cows in natural scenes, different dilation rate combinations in ASPP were discussed. When the dilation rate was (1,5,9,13), the value of mAP (IoU = 0.5) increased by 2.2%. Next, ablation comparison experiments of different optimization modules were designed, proving that different optimization modules could improve the accuracy of the model for the detection of estrus behavior in dairy cows. The C3SAB and C3DAB modules

compensated for the feature loss caused by ASPP and improved the mAP (IoU = 0.5) value by 0.4% and 0.5%, respectively. Compared with previous studies, CEBD-YOLO had the highest accuracy in natural breeding scenarios with a large number of cows, and the inference speed was better than that of the models proposed by other studies. A visual analysis of the feature extraction results of the model showed that it could identify cows that were mounting in complex environments, which explains the effectiveness of the model. In summary, the model proposed in this paper had a good effect on the detection of estrus behavior of dairy cows in natural scenes and a fast inference speed, which is suitable for real-time detection of estrus behavior of dairy cows in natural scenes and provides a judgment basis for timely detection of estrus in dairy cows.

**Author Contributions:** Conceptualization, R.W., C.Z. and R.G.; methodology, R.W., Z.G. and Q.L.; software, R.W.; validation, R.W. and H.Z.; formal analysis, R.W.; investigation, S.L. and L.F.; resources, R.W. and H.Z.; data curation, R.W. and C.Z.; writing—original draft preparation, R.W.; writing—review and editing, R.W., H.Z. and S.L.; visualization, R.W.; supervision, C.Z. and R.G.; project administration, R.G.; funding acquisition, R.G. All authors have read and agreed to the published version of the manuscript.

**Funding:** This work was supported by the National Key Research and Development Program of China (No. 2019YFE0125400) and technological innovation capacity construction of Beijing Academy of agricultural and Forestry Sciences (kjcx20220404).

**Institutional Review Board Statement:** The animal study protocol was approved by the Ethics Committee of Northwest A&F University (Approval Number: IFA-2022001).

**Data Availability Statement:** All data are presented in this article in the form of figures and tables.

**Conflicts of Interest:** The authors declare no conflict of interest.

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
