# Peer review of "Detection Method of Cow Estrus Behavior in Natural Scenes Based on Improved YOLOv5"

_agriculture, doi:10.3390/agriculture12091339_

Round 1

Reviewer 1 Report

The paper proposed a cow oestrus behaviour detection method based on the improved YOLOv5 with an accuracy of 94.3 % with a 71 Frame per second speed. A 2668 images were used for training set and 678 for training the model using K-means clustering method. The paper presents a good topic, however, the paper can be improved further as shown below:

1.     In the abstract, clear sentences show the research gap that this paper tried to solve. In other words, what is the problem of previous methods that this paper solved? also, the highest founded results.  More about the novelty of the paper.

2.     “Ablation comparison experiments with different optimization modules, different dilation rate combinations and different loss functions suggest that the optimization methods proposed in this paper can improve the accuracy of the model for detect cow oestrus events”.  These lines can be rephrased as it doesn’t show clarity of the sentences.

3.     CloU, fps, ROI, PReLu, mAP , GFLOPS, IoU, MB needs to be addressed in their full forms.

4.     Use the same terminology through the paper. Somewhere used mAP and MAP

5.     Literature review is not presented in the manuscript which validate the improved model and accuracy of YOLOv5 in this study. The authors are encouraged to present the results of previous papers in the tabular form.

6.     Kindly check the section 2.1 and the time frame, which is mentioned From September 26, 2022 to October 6, 2022.

7.     The formatting of the paper need to be improved.

8.     Nowhere mentioned why K-means to be chosen for the training the model.

9.     After section 3, authors suggested not to use direct subsection, add the text between them.

10.  Figure 3 and 10 needs to be redrawn as it lacks clarity.

11.  In Section 2.2, Check the spelling of LabelImg.

12.  Also, check subsection 3.2.5.

13.  The manuscript needs to be proofread for spelling and grammatical errors.

14.  The proposed method or novelty is missing in the introduction.

15.   The formulae performance metrics numbers are not mentioned in the text. The font size and style are different that needs to be fixed for the formulae.

16.   Less references are used. The authors are encouraged to refer to recent studies and cite them in the manuscript.

17.  The conclusion does not summarise the results obtained in the research work.

18.  This paper only splits the data into train and test, with no validation dataset. This makes the results easily overfitting on the test data.

Reviewer 2 Report

1. There is a confusion with reference numbers, please check carefully

2. In section 2.2., the period of data gathering is in the future?

3. In  section 2.2. the data enhancement process needs more explanation, and Figure 2. should be more informative (add more arrows for example. to illustrate "image flow")

4. Figure 4. is missing a separate prediction module block, although it is mentioned in the paragraph above

5. Modules in Figure 4 should be explained in the text for better readability. You mention CSP modules in the paragraph below the figure, but without notion what they do. Also, you should explain the function of modules M2, M3, M4 and P2, P3 and P4. Emphasize the motivation for having three of each (P* and M*)

6. In Section 4.1 you use abbreviation DL, I propose to use the full term instead.

7. In section 3.1 you mention CSPDarknet 53 in the text, but the context is not clear. You might also want to add reference to it in the Figure 4.

 8. Generally, avoid using abbreviations without defining them in the text. For example, DW in Figure 6 is not defined in the text.

9. You should explain, in the text, the meaning of parameter w in Figure 6.

Round 2

Reviewer 1 Report

The manuscript in the present form can be published but with minor corrections as there are few spelling mistakes in the manuscript and grammatical errors so kindly rectify them.
